# Suitability of just-in-time adaptive intervention in post-COVID-19-related symptoms: A systematic scoping review

**Gerko Schaap**[ORCID]*, **Benjamin Butt, Christina Bode**

Section of Psychology, Health and Technology, University of Twente, Enschede, The Netherlands

* g.schaap@utwente.nl

## Abstract

Patients with post-COVID-19-related symptoms require active and timely support in self-management. Just-in-time adaptive interventions (JITAI) seem promising in meeting these needs, as they aim to provide tailored interventions based on patient-centred measures. This systematic scoping review explores the suitability and examines key components of a potential JITAI in post-COVID-19 syndrome. Databases (PsycINFO, PubMed, and Scopus) were searched using terms related to post-COVID-19-related symptom clusters (fatigue and pain; respiratory problems; cognitive dysfunction; psychological problems) and to JITAI. Studies were summarised to identify potential components (interventions options, tailoring variables and decision rules), feasibility and effectiveness, and potential barriers. Out of the 341 screened records, 11 papers were included (five single-armed pilot or feasibility studies, three two-armed randomised controlled trial studies, and three observational studies). Two articles addressed fatigue or pain-related complaints, and nine addressed psychological problems. No articles about JITAI for respiratory problems or cognitive dysfunction clusters were found. Most interventions provided monitoring, education or reinforcement support, using mostly ecological momentary assessments or smartphone-based sensing. JITAIs were found to be acceptable and feasible, and seemingly effective, although evidence is limited. Given these findings, a JITAI for post-COVID-19 syndrome is promising, but needs to fit the complex, multifaceted nature of its symptoms. Future studies should assess the feasibility of machine learning to accurately predict when to execute timely interventions.

## Author summary

Patients with post-COVID-19 syndrome (PCS) experience a lot of various, continuously fluctuating symptoms. Certain mobile health technologies, such as just-in-time adaptive intervention (JITAI), may help in dealing with these symptoms, and

**Data availability statement:** All data required to replicate the results of this study are included in S1 Table.

**Funding:** GS was financially supported by the foundation 'Friends of MST' (Dutch: stichting Vrienden van MST; https://www.samenvoormst.nl/?locale=en), which through donations supports projects and initiatives to improve healthcare and health practices within and beyond Twente, the Netherlands. BB and CB have no funding to declare. The funders had no role in study design, data collection and analysis, decision to publish, or preparation of the manuscript.

**Competing interests:** The authors have declared that no competing interests exist.

have been utilised in other health conditions. In this study, we investigate to what extent it would be possible to develop a JITAI for post-COVID-19 and what that would look like based on 11 existing JITAI-related articles. We found evidence for JITAI being helpful to deal with psychological problems (such as depression) and to a lesser extent with aspects of fatigue, using sensing technologies or user-reported variables. Our findings suggest that JITAI have potential to help PCS patients. However, more research is necessary to develop a JITAI to help patients deal with other symptoms (such as cognitive problems and breathing difficulties), as well as on how to predict when interventions can occur at the right time for fatigue. These findings may also be helpful to develop interventions in other disorders, such as chronic fatigue syndrome.

## Introduction

COVID-19 has had an enormous lasting effect on the world since its emergence. Among those effects are the persisting sequelae COVID-19 survivors experience that are commonly referred to as long COVID, or post-COVID-19 syndrome. Post-COVID-19 syndrome is defined as "signs and symptoms that develop during or after an infection consistent with COVID-19, continue for more than 12 weeks and are not explained by an alternative diagnosis" [1]. This syndrome occurs on estimation in one in eight COVID-19 survivors [2].

Post-COVID-19-related symptoms can vary between and within post-COVID-19 patients. To illustrate, over 200 different symptoms associated with different organ systems have been reported [3]. However, the most commonly reported symptoms can be summarised in various symptom clusters, as for example by (co-)occurrence and severity [4]. 1) Persistent fatigue with bodily pain are among the most commonly reported symptoms [5], and can encompass both physical and mental fatigue [6], but also post-exertional malaise (PEM) [7]. This cluster resembles psychosomatic syndromes such as chronic fatigue syndrome and fibromyalgia. 2) Respiratory problems can include (exertional) breathlessness, cough, wheezing, and chest discomfort [4]. 3) Cognitive dysfunction can include various neurological complaints, such as attention disorders, memory problems, and confusion or disorientation, sometimes referred to as brain fog [4,5]. Besides these three clusters, 4) psychological problems including depression, anxiety, insomnia, and post-traumatic stress disorder are commonly reported [8,9]. These symptoms clusters often co-occur in various levels of severity [10] and positively associate over time [11], requiring complex, multifaceted interventions for improvement and management.

To consider treatment and management of post-COVID-19 symptoms, diagnosis starts by screening for presence and severity of symptoms through clinical questionnaires and, depending on symptoms and functioning, can also include assessing signs through clinical, radiological and biochemical assessment. Imaging can include high resolution CT scans for pulmonary state ("Possible image findings include ground glass opacification, consolidation, crazy paving, vacuole sign, pulmonary

nodules, lobar pneumonia, tractional bronchiectasis, vascular thickening, lung cavitations and fibrosis, predominantly at the peripheral/subpleural locations" [12] (p4)), chest X-rays and cardiac MRI for cardiovascular problems, and MRI scans to exclude alternative diagnosis or worsening of neurocognitive disorders. Biochemical research can include investigating clotting problems (e.g., increased levels of antiplasmin), inflammation biomarkers (e.g., persistence of interleukin-6 and tumour necrosis factor-alpha), metabolic markers (blood glucose, HbA1c levels), and vitamin deficiency (e.g., vitamins B12 and D) [10,12]. However, research on best practices in assessment and monitoring is still ongoing; for an overview, see Yan et al. [12]. Based on symptoms and signs, patients may get referred to healthcare providers for treatment.

To date, no curative treatment for post-COVID-19 syndrome exists, and treatment and clinical management aim towards rehabilitation and symptom alleviation [3]. While healthcare providers, such as physicians, physiotherapists, occupational therapists, and psychotherapists, play a crucial role in treatment, patients need to deal with their illness in daily life using self-management. Some post-COVID-19 symptoms, such as autonomic dysfunction, allergic responses (e.g., rashes, loss of taste and smell) and hair loss are relatively static (i.e., do not change throughout the day) and can be treated solely using biomedical intervention. Yet others, such as the four symptom clusters, tend to be at least partly time-varying and sensitive to context (e.g., exerting activities) [6,11,13]. Consequently, patients need to actively manage their symptoms themselves, for example through pacing strategies (i.e., carefully managing energy in order to not overexert) [14]. However, certain interventions can support patients in their self-management.

eHealth can be promising for such interventions in supporting post-COVID-19-related symptom management. Firstly, eHealth interventions can either support or replace traditional (face-to-face) treatments, relieving the strain on healthcare providers whilst being cost-effective. Secondly, eHealth interventions are scalable to larger target populations and have a low threshold, allowing to include patients who currently have little to no contact with healthcare providers. Thirdly, the diverse nature of post-COVID-19 syndrome necessitates multidisciplinary and tailored approaches, which eHealth interventions can provide [15]. A recent scoping review identified that eHealth interventions for post-COVID-19 syndrome already exist and preliminary evidence suggest that these are beneficial for symptom clusters such as fatigue and psychological problems [16]. However, so far, no interventions are known that support post-COVID-19 patients and intervene in real-time when they need help with their symptoms the most. Such interventions may be helpful to prevent, for example, PEM [17,18]. Moreover, post-COVID-19 symptoms seem to be at least partly time- or context-specific [6,11,19], suggesting that timely support in self-management might prevent or ameliorate these complaints. One type of intervention that can account for that is Just-in-Time Adaptive Intervention (JITAI).

JITAIs are mobile health (mHealth) interventions that dynamically deliver tailored support at the right moment using real-time data to support self-management [20]. These systems are designed to help individuals, based on their own input, with the self-regulation of prespecified health behaviours, and aim to intervene at the exact moment of need, typically through behaviour change techniques. For example, for supporting smoking cessation, an intervening message on coping skills can get delivered when stress becomes elevated and craving increases. Advantages of JITAIs include that they can be delivered remotely and require different resources than traditional face-to-face counselling. This makes them promising for improving access to (mental) healthcare and supporting individuals in managing chronic health conditions. Additionally, because they are dynamically tailored to the user and situation, intervention adherence and retention should be fully supported [20]. Despite this potential, there is currently a lack of literature on the effectiveness of JITAIs [21], especially for post-COVID-19-related symptom clusters. Furthermore, the development of JITAIs is resource-intensive and methodologically and logistically challenging due to how the components work [22,23]. To consider the suitability of (developing) a potential post-COVID-JITAI, the key components need to be defined.

Following previous reviews [20,21,24], four key components of JITAI design can be specified: intervention options, tailoring variables, decision rules, and decision points (see Fig 1). 1) Intervention options are the interventions potentially provided at a certain decision point. They can differ by type (e.g., support, information, or feedback), source (technological or human), delivery system (e.g., push message, email or phone call) and intensity, but need to fit the right contextual and

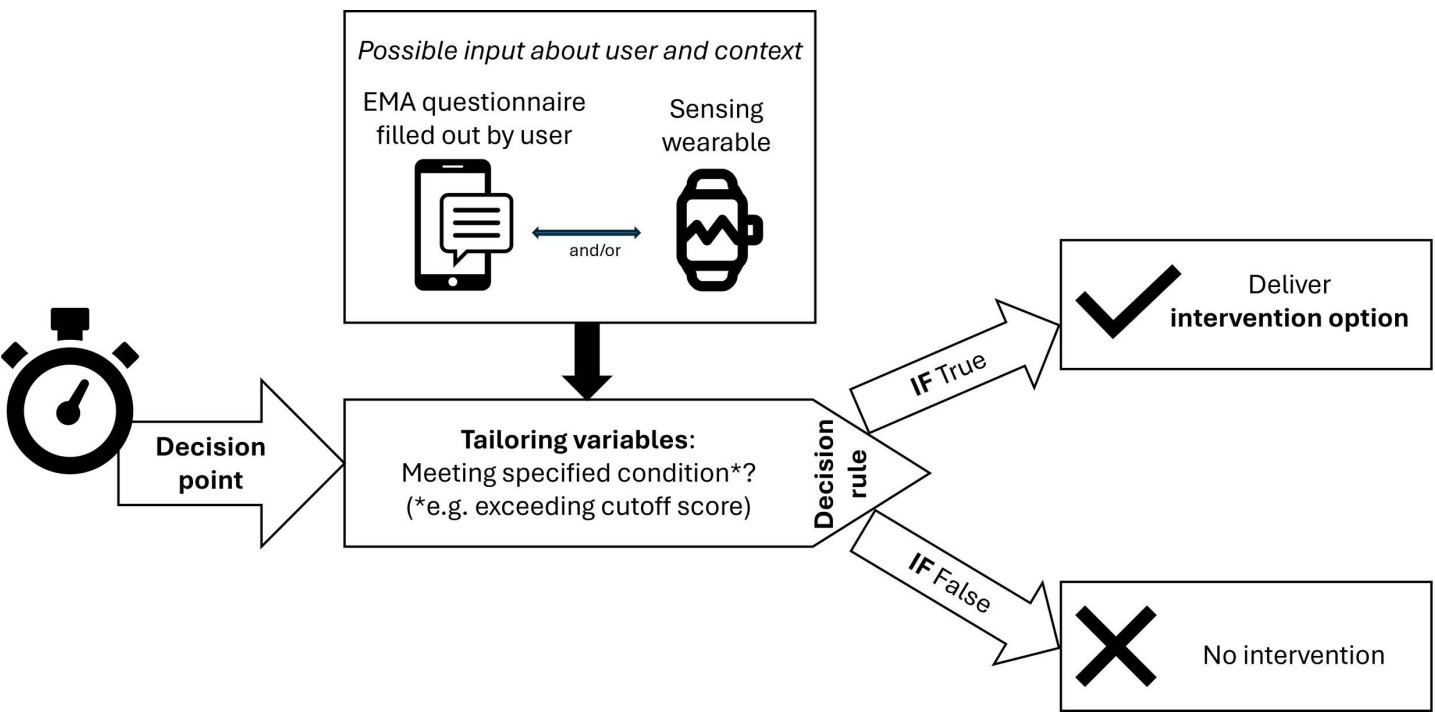

**Fig 1. Conceptual overview of key components of just-in-time adaptive interventions.**

temporal conditions, and hence require tailoring. 2) Adaptive tailoring variables provide such information. These variables can be collected passively (e.g., using wearables such as accelerometers or heart rate trackers, or via integrated electronic health records), actively using ecological momentary assessment (EMA; e.g., patient-reported mood or symptom levels) or both [25]. 3) Decision rules are the operationalising algorithms that specify what intervention options should be offered, under which conditions, and for whom. Simplified, these rules can be 'IF (fatigue score ≥ [cutoff]) THEN (provide advice), IF ELSE (do nothing)'. These decision-making algorithms can be statically prespecified by researchers, i.e., based on theory or prior evidence, or dynamically computed to fit the patient, e.g., by machine learning models. These decision rules trigger at a 4) decision point, which specifies when an intervention option might be delivered. This may for example be directly after the measurement, a random prompt, or at prespecified or random time windows (e.g., every minute or twice daily).

While evidence seem to suggest that JITAIs are effective in supporting self-management [21], it is unclear whether these advanced mHealth interventions would be suitable, feasible and effective for post-COVID-19-related symptom clusters, how such a JITAI would look like, and what would be any potential barriers (e.g., in user engagement) that need to be overcome. As decision points are informed by the other components, they are outside of this study's scope of interest. This research could help inform decisions in the development of JITAI that can support patients with post-COVID-19-related symptoms. Thus, the objective of the current study was to scope out and identify elements of interest using the following research questions:

*RQ1.How are JITAI components (a. interventions options, b. tailoring variables and c. decision rules) applied in JITAIs related to post-COVID-19 symptom clusters?*

*RQ2.What can be said about the feasibility, effectiveness and potential barriers of JITAIs related to post-COVID-19 symptom clusters?*

PLOS Digital Health

## Methods

### Design

The systematic scoping methodological framework proposed by Arksey and O'Malley [26] was followed. The process involved: (1) identifying research questions; (2) identifying relevant studies; (3) selecting studies; (4) charting the data; and (5) collating, summarising, and reporting the results. The study is reported conforming the Preferred Reporting Items for Systematic reviews and Meta-Analyses extension for Scoping Reviews (PRISMA-ScR) checklist [27] (S1 Checklist). This study was registered at Open Science Framework Registries: https://osf.io/kgecs.

### Search strategy

Three electronic databases (PubMed, Scopus and PsycINFO) were initially queried. The search terms were composed of (1) synonyms for JITAI and (2) symptom clusters, separated by Boolean operators. Specifically, the string encompassed ("JITAI" OR "Just-in-time adaptive intervention" OR "just-in-time" OR "adaptive intervention" OR "dynamic intervention") AND ("long COVID" OR "fatigue" OR "chronic fatigue syndrome" OR "bodily pain" OR "pain" OR "myalgia" OR "cognitive problems" OR "attention disorder" OR "memory impairment" OR "respiratory problems" OR "lung disease" OR "depression" OR "anxiety" OR "insomnia"). The most recent search was performed on 5 April 2024, selecting only peer-reviewed journal articles. Afterwards, a manual snowball search through reference lists of selected studies was performed to identify other potentially relevant papers, including grey literature and book chapters. Additionally, the database ACM Digital Library was searched for relevant conference proceedings on 5 April 2024.

### Study selection

Studies were included using the following criteria: (a) addressing at least one symptom of the specified symptom clusters using JITAI; (b) available as full-text in English or German; and (c) published since January 1, 2015. Quantitative, qualitative, and mixed-method study designs were considered in the form of any type of experimental (including randomised and non-randomised controlled trials, and feasibility, pilot or usability studies) or observational setup. As the study aim was to explore JITAI component options for a potential JITAI in post-COVID-19, the studies did not necessarily have to be about evaluating JITAI specifically. Study protocols and reviews were excluded.

Studies were exported to Rayyan [28], after which duplicates were omitted. Titles and abstracts were screened against the eligibility criteria, followed by full-text evaluation, by the second author (BB). This selection was reviewed by the first and last author (GS and CB).

### Data charting and synthesis

Two authors (GS and BB) extracted data from the selected articles into tables. Information was categorised based on the research questions: type of symptom cluster, interventions options, tailoring variables, decision rules, information on effectiveness and feasibility, and discussed (potential) barriers. Additionally, author, publication year, country, type of study design, and study population were retrieved. One author (GS) synthesised the findings by deductive themes aligning with the research questions.

## Results

The initial search identified 341 studies, including 127 duplicates that were omitted. Of these 214 studies, 161 were excluded for not meeting the eligibility criteria, leaving 53 records for full-text review. Finally, 11 articles met the eligibility criteria (Fig 2). Snowball searches and searching ACM Digital Library did not result in eligible records. Included study designs varied: five single-armed feasibility or pilot studies, three two-armed randomised controlled trial (RCT) studies, and three observational studies.

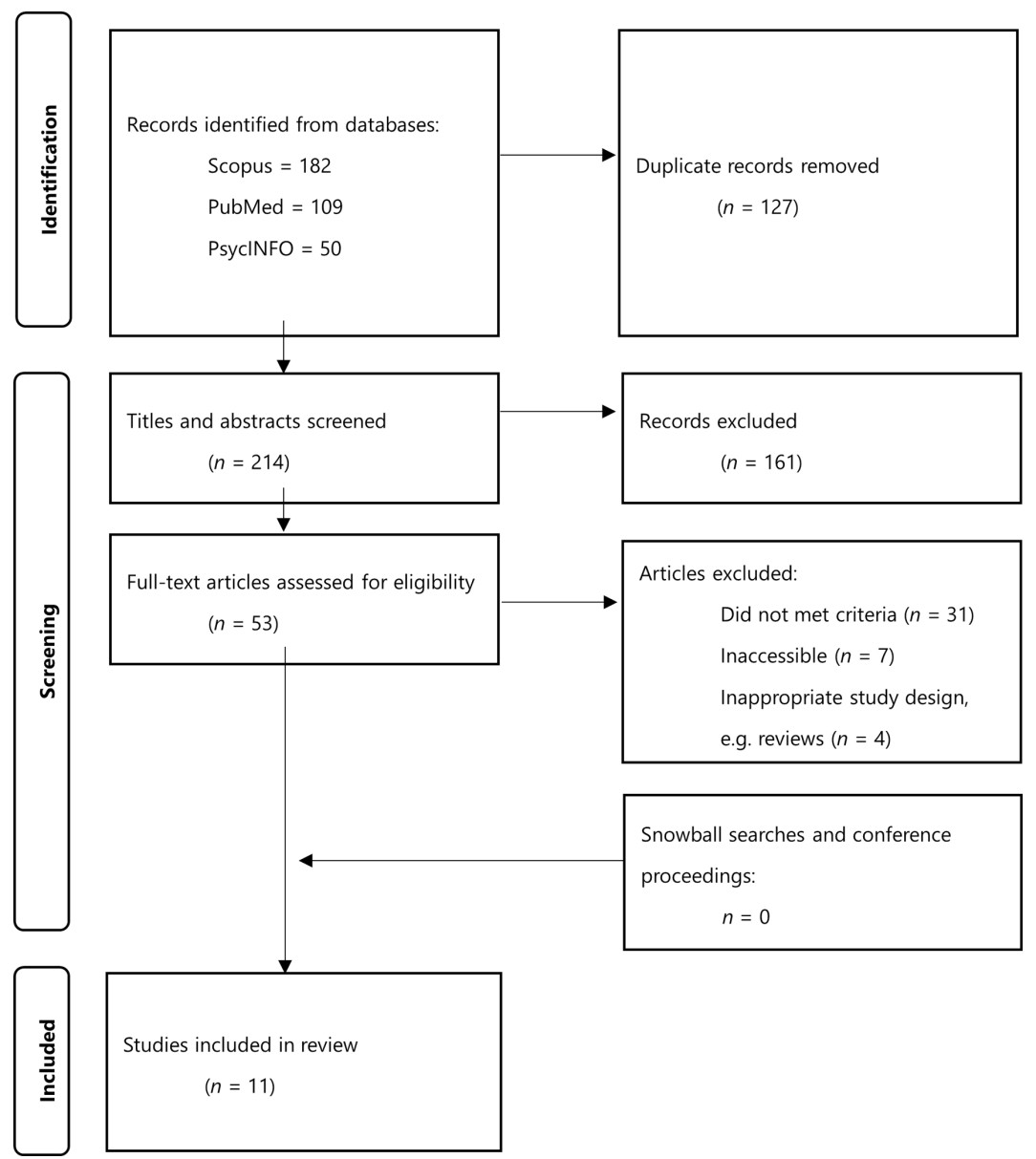

**Fig 2. Preferred Reporting Items for Systematic Reviews and Meta-Analyses (PRISMA) flow-chart of included studies.**

Included study characteristics are presented in Table 1. Most ($n=9$) studies were executed in the United States. Populations differed significantly between studies, aligning with the heterogeneous aims, varying from adolescents and adults of non-clinical populations to individuals with spinal cord injuries. A summary of JITAI key components, effectiveness, and barriers per symptom cluster is presented in Table 2. Data per study can be found in S1 Table.

Two articles addressed fatigue or pain-related complaints, and nine addressed psychological problems. No articles about JITAI for respiratory problems or cognitive dysfunction clusters were found.

**Table 1. Characteristics of included studies per post-COVID-19-related symptom cluster.**

| Authors (year) | Country | Name of JITAI | Target outcome(s) | Population | Sample size | Study design | Study duration |
|---|---|---|---|---|---|---|---|
| *Fatigue and pain symptom cluster* | | | | | | | |
| Azizoddin et al. (2024) [29] | United States | STAMP+CBT | Improve pain management | Adults with advanced cancer and chronic pain in palliative care | 15 | Single-armed feasibility study | 2–4 weeks |
| Hiremath et al. (2019) [30] | United States | PHIRE | Promote physical activity | Adults with spinal cord injury, using a manual wheelchair as primary means of mobility | 20 | Single-armed feasibility study | 6 weeks |
| *Psychological problems symptom cluster* | | | | | | | |
| Bell et al. (2023) [31] | Australia | Mello | Reduce repetitive negative thinking, anxiety, and depression | Adolescents and young adults with clinical levels of depression and anxiety, and heightened repetitive negative thinking | 55 | Two-armed pilot RCT JITAI: $n=29$ Control: $n=26$ (non-active) | 6 weeks |
| Beltzer et al. (2022) [32] | United States | N/A | Develop algorithms for emotional regulation strategies in social anxiety | Undiagnosed adults with moderate to severe social anxiety | 110 | Observational EMA | 5 weeks |
| Carlozzi et al. (2022) [33] | United States | Not reported | Reduce caregiver strain, anxiety, depression, and sleep-related impairment | Care partners of persons with spinal cord injury, Huntington disease or hematopoietic cell transplantation | 70 | Two-armed RCT 3 months JITAI: $n=36$ Control: $n=34$ (only monitoring) | 3 months |
| Jacobson and Chung (2020) [34] | United States | N/A | Predict changes in depressed mood | Undiagnosed young adults with moderate to very severe depression severity | 31 | Observational EMA | 1 week |
| Pulantara et al. (2018) [35] | United States | iREST | Improve sleep behaviours and sleep quality | Military personnel or veterans with persistent sleep disturbances | 19 | Single-armed usability study | 6 weeks |
| Pulantara et al. (2018) [36] | United States | iREST | Improve sleep behaviours and sleep quality | Military personnel or veterans with persistent sleep disturbances | 27 | Single-armed pilot study | 5 weeks |
| Ren et al. (2023) [37] | United States | N/A | Predict elevated negative affect | Adolescents with low to high depression severity, including anhedonia | 22 | Observational EMA | 5 weeks |
| Wahle et al. (2016) [38] | Switzerland | MOSS | Reduce depression severity; Predict depressive state | Adults considered to be clinically depressed | 36 with at least 2 weeks adherence; 28 with at least 4 weeks adherence; 12 with pre- and post-measurements | Single-armed pilot study | At least 2 weeks, up to 9 months |
| Wang and Miller (2023) [39] | United States | Not reported | Reduce rumination | Adults diagnosed with and actively treated for clinical depression | 18 | Two-armed pilot RCT 35 days JITAI: $n=9$ Control: $n=9$ (data collection only) | 35 days |

*Notes.* EMA = ecological momentary assessment; JITAI = just-in-time adaptive intervention (group); N/A = not applicable; RCT = randomised controlled trial.

**Table 2. Summary of JITAI key components, evaluation, and barriers per post-COVID-19-related symptom cluster.**

| Symptom cluster | Intervention option | Tailoring option | Decision rules | Effectiveness/ feasibility | Barriers |
|---|---|---|---|---|---|
| Fatigue and pain (*n* = 2) | Supportive messages with information on pain or reinforcement messages for and feedback on physical activity goal attainment | Based on symptom-specific EMA or passively measured physical activity | Based on static (pre-defined) algorithms; activated daily or when continuous moderate-to-vigorous physical activity was detected | Both JITAI were deemed highly acceptable and feasible; limited evidence for effectiveness | Minor technical issues and minor issues with participant burden (e.g., too much information provided; lack of control) were encountered |
| Respiratory problems | No search hits (*n* = 0) | | | | |
| Cognitive dysfunction | No search hits (*n* = 0) | | | | |
| Psychological problems (*n* = 9) | Mostly delivery of CBT (micro-)interventions or other educational content to improve mood, well-being or sleep, or reduce symptoms | Mostly based on symptom-specific or symptom-related EMA. Passive accelerometer-based and smartphone-based sensing (e.g., physical activity, location and social activity) were also used | Primarily based on static (predefined) algorithms to push personalised content. One study showed that dynamic adaptation can be applied | JITAI were found to be acceptable and feasible; limited evidence for effectiveness. EMA studies showed that psychological outcomes can be predicted using machine learning models | Some studies reported technical issues. Motivation for adherence and compliance can be an issue with EMA tailoring; passive sensing could reduce this barrier |

*Notes.* Summary based on 11 articles. CBT = Cognitive Behaviour Therapy; EMA = Ecological Momentary Assessment

### Fatigue and pain-related JITAI components

The included articles included two JITAIs; one focusing on pain management [29] and one on physical activity improvement [30], which might contribute to fatigue reduction.

The JITAIs provided interventions that coached participants in their self-management, namely education for pain management based on cognitive behaviour therapy (CBT) for pain [29] and reinforcing physical exercise and exercise goal attainment [30]. These interventions were initiated using static algorithms. For pain management, messages were always sent daily, but the contents were tailored using EMA of symptoms and symptom-related behaviour (e.g., pain catastrophising) with predetermined cutoff scores. For physical activity, encouraging messages were sent when moderate-to-vigorous physical activity was detected using machine learning-instigated estimations and objective measures. Daily goal attainment was checked again personalised goals. Both JITAI were found to be highly acceptable and feasible, with few, minor barriers being reported.

### Psychological problems-related JITAI components

Six of the included articles reported on five JITAIs for anxiety, disordered mood and thoughts, and sleep disturbances [31,33,35,36,38,39], while three other articles provided useful information about JITAI components for psychological symptoms [32,34,37].

The majority (5 out of 6) of studies provided interventions based on CBT approaches, including (self-regulation) skill development, as well as monitoring [33,35,36,38,39]. Tailoring measurements mostly relied on EMA for mood and symptoms [31,33,39] or health behaviours [35,36]. For passive measurements, one study used an accelerometer for physical activity and sleep in addition to EMA [33], while one study used smartphone sensing data of time, location (GPS and Wi-Fi), and smartphone-based social activity (e.g., texting and calling) [38]. Primarily (5 out of 6 studies), static algorithms were deployed as decision rules. Wahle et al. applied a hybrid system after an initially static algorithm, to dynamically tailor the probability of behaviour activation recommendations based on historical user data [38]. In contrast, the JITAI of Pulantara et al. required manual triggering of the intervention by clinicians [35,36].

JITAIs were evaluated as highly feasible, acceptable and usable. Few barriers were reported, although some JITAI experienced technical issues that needed to be overcome [35], and motivation and adherence being potential limitations with EMA [31,38] or with using wearables [33]. Preliminary effectiveness was found to be moderate to good, although all studies seemed to have been underpowered. The RCTs [31,33,39] observed outcomes significantly favourable for the JITAI groups, with reported effect sizes being medium to large (Cohen's *d* between 0.5 and 0.87) [31] and large (Cohen's *d* of 1.84 and 2.5) [39]. The non-randomised studies likewise observed significant improvements in outcomes [36,38].

Finally, two observational studies showed how smartphone sensing data could be used to predict depressive states using machine learning models [34,37]. Moreover, Beltzer et al. showed that the effectiveness of emotion regulation strategies against anxiety could be predicted and can be tailored to individuals [32].

## Discussion

This scoping review examined a heterogeneous set of JITAIs, targeting a variety of fatigue, pain, and psychological symptoms in various patient and non-patient populations. Interestingly, no JITAI or JITAI-related (observational) studies could be identified that targeted respiratory problems or cognitive dysfunction. Important insights into key components (a. interventions options, b. tailoring variables and c. decision rules) for a potential post-COVID-JITAI were charted, showing various applications. Additionally, JITAI and related (machine learning) algorithms seem to be generally acceptable, feasible, and effective, although evidence is limited. Few barriers were discussed in the included studies, but results indicated that stakeholder analysis (e.g., of patient and healthcare provider expectations) is important, and (minor) technical issues should be expected.

While these findings suggest promising implications for a post-COVID-JITAI, the suitability is still questionable. As seen in the included studies, as well in most JITAIs, most intervention options aim towards behaviour activation of health behaviours [21], such as encouraging to continue physical exercise. However, one major challenge is that encouraging activity and exertion can strongly backfire in many post-COVID-19 patients due to PEM – a post-COVID-19 JITAI might actually need to advise patients not to do anything at the moment to avoid PEM [3,10]. Current rehabilitation strategies aim more towards pacing and energy management over reconditioning via (e.g.,) graded exercise [14,40]. As fatigue and PEM are among the most commonly reported and most severely experienced symptoms, they need to be appropriately addressed in a JITAI format. Help with energy management is also one of the most favoured eHealth intervention desires of post-COVID-19 patients [41]. A JITAI may partly aid in this via activity monitoring (e.g., based on physical and cognitive activity of the previous hour [19]) and feedback to provide insights in daily life exertions to help patients structure their days. Besides the fatigue cluster, studies from the psychological problems cluster show that CBT-based education and skill development may be beneficial for helping with emotion regulation. Explorative observational studies suggest that improved mood and reduced psychopathology may also lead to improved somatic post-COVID-19 symptoms [11,42]. While intervention options need to be carefully developed, a JITAI for post-COVID-19 patients may be possible, albeit with sensible tailoring.

Tailoring variables of the selected JITAIs used both EMA and passive, objective measures. Multiple included studies showed that smartphone-based sensing data of physical and social activities provide good proxy measures for psychopathologies, and that accelerometers can help with physical activity tracking. Somatic symptoms – especially those without proxies or (accessible) biomarkers – still may require EMA. In particular fatigue tailoring is challenging given its complex, multifaceted, subjective and fluctuating nature [43]. As both false positives and false negatives (or over- and underestimation) of severe fatigue can lead to the wrong triggering decision, accurate identification is crucial. Schneider et al. [44] argue for an adaptive assessment via fatigue questionnaire items that dynamically tailor ("by individual and time of day") the cutoff scores with fewer EMA questions [44] (p770). Furthermore, not only can artificial intelligence models help improve the understanding of fatigue in general [43], machine learning models can be leveraged to also incorporate wearable biosensors, resulting in more accurate, multimodal predictions [45]. For example, individually tailored fatigue EMA

may be supplemented with skin conduction measurements. A post-COVID-JITAI should also be able to tailor interventions to multiple, potentially interrelated symptoms (e.g., depressed mood and breathlessness), signals (e.g., skin conduction and heart rate), and behavioural and environmental factors (e.g., activity, background noise, and light intensity). The feasibility of these complex predictions should be tested first similar to the observational studies included in our findings [32,34,37].

Similarly, future steps are needed to develop reliable, dynamic decision rules based on these symptoms and signals. Most of the included studies featured static, partially automated systems, which is common for JITAI. However, full automation requires less work for users and healthcare providers [46]. Only after identifying the correct, accurate predictors can these decision rules be designed.

Interestingly, no JITAIs for the cognitive dysfunction and respiratory problems clusters were found. While specific interventions for cognitive dysfunction (e.g., disorientation) may not be necessary, a post-COVID-JITAI should be able to account for these states. For respiratory problems, mHealth interventions for pulmonary rehabilitation exists, such as providing breathing techniques [47]. While it may not be necessary to trigger these 'just-in-time', they can be incorporated into a post-COVID-JITAI for those with respiratory problems as, for example, a daily exercise as well as monitoring. That way, post-COVID-19 patients can have a single system aiding in symptom management adapted to their personal needs [41].

While the feasibility and acceptability should be studied further, JITAIs are (relatively) easily embraced by patients. Minor barriers point towards careful investigations with stakeholders (patients and healthcare providers), but they seem to fit well with participants' expectations. Nevertheless, the evidence for effectiveness requires more research in larger samples, as is common for JITAI RCTs [21].

This study encompasses a scoping overview of JITAIs for illnesses with post-COVID-19-related symptoms using a rigorous methodology [26]. The findings should however be interpreted with caution. The search query was relatively simple and could have been optimised by expanding on the included symptom-related terms. Snowball searching for related JITAIs was executed to make up for this. On a related note, there currently is no consensus on definitions of JITAI. While the query included multiple possible related terms, others could have been missed. Ecological momentary interventions were excluded from the query due to the scope and time, but might also have been insightful as non-automated or partially automated intervention systems. Simarly, while over 200 different post-COVID-19 symptoms can be defined [3], by choice the scope of this study was on the most commonly reported, potentially fluctuating symptoms. This may not align with the needs and preferences of patients [41], necessitating follow-up studies.

## Conclusion

The aim of this scoping review was to explore the suitability of a JITAI for post-COVID-19 syndrome. Based on JITAIs for post-COVID-19-related symptoms key components were discussed, and found to be promising, but many questions remain. A post-COVID-JITAI could help handle the fluctuating nature of symptoms and support in the various difficulties in self-management. Future studies should look into the feasibility of machine learning to accurately predict symptom severity in complex, multifaceted and interrelated mess of post-COVID-19 syndrome to trigger psychoeducation, cognitive behaviour therapy and monitoring and early warnings for post-exertional malaise. This information may also benefit timely interventions in other complex (psycho)somatic disorders such as fibromyalgia and myalgic encephalomyelitis/chronic fatigue syndrome.

## Supporting information

**S1 Checklist.** PRISMA-ScR checklist.
(PDF)

**S1 Table.** Data of included studies. Extracted JITAI key components, evaluation, and barriers of included articles.
(DOCX)

## Author contributions

**Conceptualization:** Gerko Schaap, Benjamin Butt, Christina Bode.

**Formal analysis:** Gerko Schaap.

**Investigation:** Gerko Schaap, Benjamin Butt.

**Supervision:** Christina Bode.

**Writing – original draft:** Gerko Schaap.

**Writing – review & editing:** Gerko Schaap, Benjamin Butt, Christina Bode.

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
