## [Decision Letter · Decision Letter 0]

18 Mar 2025

Response to Reviewers
Revised Manuscript with Track Changes
Manuscript
**Journal Requirements:**

i. Please clarify all sources of funding (financial or material support) for your study. List the grants (with grant number) or organizations (with url) that supported your study, including funding received from your institution. 

ii. State the initials, alongside each funding source, of each author to receive each grant.

iii. State what role the funders took in the study. If the funders had no role in your study, please state: “The funders had no role in study design, data collection and analysis, decision to publish, or preparation of the manuscript.”

iv. If any authors received a salary from any of your funders, please state which authors and which funders.

2. Please make sure the funding information on the submission form matches your financial disclosure statement. Please indicate by return the full and correct funding information for your study and confirm the order in which funding contributions should appear. Please be sure to indicate whether the funders played any role in the study design, data collection and analysis, decision to publish, or preparation of the manuscript.

3. Please send separate figure files in .tif or .eps format. Also, remove the figures from your manuscript file but keep the legends.

4. We have noticed that you have uploaded Supporting Information files, but you have not included a list of legends. Please add a full list of legends for your Supporting Information files after the references list. 

5. We notice that your supplementary Table A1 are included in the manuscript file. Please remove them and upload them with the file type 'Supporting Information'. Please ensure that each Supporting Information file has a legend listed in the manuscript after the references list.

6. We note that your Data Availability Statement is currently as follows: “N/A (however, data from studies included in this systematic scoping review is included as an appendix)” 

**Additional Editor Comments (if provided):**
**Reviewers' Comments:**

**Comments to the Author**

1. Does this manuscript meet PLOS Digital Health’s publication criteria?

Reviewer #1: Yes

Reviewer #2: Yes

2. Has the statistical analysis been performed appropriately and rigorously?

Reviewer #1: Yes

Reviewer #2: Yes

3. Have the authors made all data underlying the findings in their manuscript fully available (please refer to the Data Availability Statement at the start of the manuscript PDF file)?

Reviewer #1: Yes

Reviewer #2: Yes

4. Is the manuscript presented in an intelligible fashion and written in standard English?

Reviewer #1: Yes

Reviewer #2: Yes

Reviewer #1: The statistical analysis has been provided appropriately. Author has provided all the data underlying the findings and is written in standard english. The manuscript can be accepted for publication with minor revisions.

Reviewer #2: COVID-19 was and has been a worldwide systemic shock to the healthcare system. Curiously what has emerged has been a post COVID-19 syndrome which has a myriad of symptoms and signs, is difficult to diagnose and requires adaptive strategies to enhance diagnosis, treatment and generally manage these often-difficult patients. Just in time adaptive interventions is a strategy allowing rapid course correction in the approach to patients. In this scoping review authors screened 341 papers and identified 11 which were suitable for analysis through a “just in time lens.” The search terms utilized included symptom clusters of fatigue and pain, respiratory problems, cognitive dysfunction and psychological problems. The end points examined were those related to just in time interventions including intervention options, tailoring variables and decision rules. All in all, the authors found that in the limited number of papers identified - 11 - two articles addressed fatigue or pain and nine address psychological problems. Curiously no papers were found which addressed more primary medical issues such as respiratory complications or cognitive dysfunction. The authors conclude that just in time strategies are acceptable and feasible though with limited evidence supporting this plausible conclusion, and further go on to endorse it just in time approach when needed for rapid decision making. While this review covers much ground there are several issues which could be addressed to strengthen the paper and are discussed below.

While the review has covered and states up front “200 symptoms have been described in post COVID syndrome,” the authors then go on to examine a limited number of these. In light of the fact that out of 340 papers only 11 are presented and overall we are dealing in this paper with limited number of data points It would be useful to include a table, beyond the reference of the symptoms. While authors or others may say this is beyond the scope of the review, in light of the limited data in the review this would be a useful addition for the reader to get a better feel for the symptoms involved. If this is viewed as being too extensive perhaps the top 100 or 50 symptoms could be provided this would be a useful addition to the reader and would increase the value of the paper for a broader readership.

Relatedly while the paper focuses on symptom complexes it would be useful to address and provide a table of other signs and lab or imaging abnormalities in post COVID-19 syndrome. While one may say this is beyond the scope of the paper and relates to other reviews, having things immediately present while one reads a review like this enhances the insight and broad perspective and value for a wide readership. This should be strongly considered.

Similar suggestion applies to just in time interventions. It would be useful to add a figure diagramming how a just in time intervention works. While the authors have nicely provided a sentence or two related to smoking cessation and more general figure would enhance the value for a broad readership. Relatedly, they could bring in in that figure the key components related to intervention options tailoring variables decision rules and decision points this would make it more interpretable to others less familiar with this technology.

Definition and working understanding of just in time interventions. It is only at the end of the paper that in the discussion it is called out that there's no consensus on definitions of just in time interventions. While the reviewer understands that this is a review on just in time interventions for specific COVID related situation a bit of a deeper discussion about what just in time interventions are, what is their scope? how do they work? what are the pros and cons? would be useful to enhance the value for the broader readership. This should be placed both up front as well as in the discussion.

Similarly adding to the figure images related to wearables and how information enters and exits a JITAI strategy would be useful

It would be very useful to include a table of abbreviations. While there are many complex terms introduced and abbreviations are then cited, as this article is long one often needs to go back to identify what these abbreviations are. For those readers unfamiliar with this area of work finding things such as PEM and others in one location in a table would be useful.

It would be very useful to address the utility of just in time interventions for more substantive signs and laboratory abnormalities associated with post COVID syndrome. While it is understood that this is a vague syndrome and the paper addresses symptom complexes the utility of this with an eye to the future, i.e. aimed at data provided by periodic lab tests or wearables even more dynamically, could have impact on more organic involvements in post COVID syndrome. The authors could at least provide their vision as to how this might be useful - to again add more data or perspective to this review.

**Do you want your identity to be public for this peer review?** For information about this choice, including consent withdrawal, please see our Privacy Policy

Reviewer #1: **Yes: ** Ankit Agarwal

Reviewer #2: **Yes: ** Marvin J. Slepian

**Figure resubmission:****Reproducibility:** To enhance the reproducibility of your results, we recommend that authors of applicable studies deposit laboratory protocols in protocols.io, where a protocol can be assigned its own identifier (DOI) such that it can be cited independently in the future. Additionally, PLOS ONE offers an option to publish peer-reviewed clinical study protocols. Read more information on sharing protocols at https://plos.org/protocols?utm_medium=editorial-email&utm_source=authorletters&utm_campaign=protocols

---

## [Decision Letter · Decision Letter 1]

2 May 2025

Suitability of just-in-time adaptive intervention in post-COVID-19-related symptoms: A systematic scoping review

PDIG-D-25-00095R1

Dear Mr Schaap,

We are pleased to inform you that your manuscript 'Suitability of just-in-time adaptive intervention in post-COVID-19-related symptoms: A systematic scoping review' has been provisionally accepted for publication in PLOS Digital Health.

Best regards,

Haleh Ayatollahi

Section Editor

PLOS Digital Health

**Additional Editor Comments (if provided):**

**Reviewer Comments (if any, and for reference):**

Reviewer's Responses to Questions

**Comments to the Author**

Reviewer #1: All comments have been addressed

Reviewer #2: All comments have been addressed

publication criteria?

Reviewer #1: Yes

Reviewer #2: Yes

3. Has the statistical analysis been performed appropriately and rigorously?

Reviewer #1: Yes

Reviewer #2: Yes

4. Have the authors made all data underlying the findings in their manuscript fully available (please refer to the Data Availability Statement at the start of the manuscript PDF file)?

Reviewer #1: Yes

Reviewer #2: Yes

5. Is the manuscript presented in an intelligible fashion and written in standard English?

Reviewer #1: Yes

Reviewer #2: Yes

Reviewer #1: ok to accept.

Reviewer #2: The authors ahve nicely addressed and/or incorporated suggestions to strengthen the paper. It is now much clearer and stronger.

**Do you want your identity to be public for this peer review?** For information about this choice, including consent withdrawal, please see our Privacy Policy

Reviewer #1: **Yes: ** Ankit Agarwal

Reviewer #2: **Yes: ** Marvin J. Slepian
